# First Insights into Ploidy and Genome Size Estimation in *Choerospondias axillaris* (Roxb.) B.L.Burtt & A.W.Hill (Anacardiaceae) Using Flow Cytometry and Genome Survey Sequencing

**DOI:** 10.3390/plants14193094

**Published:** 2025-10-07

**Authors:** Fangdi Li, Zhuolong Shen, Tianhe Zhang, Xiaoge Gao, Huashan Ling, Hequn Gu, Zhigao Liu, Jiyan Liu, Chaokai Lin, Qirong Guo

**Affiliations:** 1Co-Innovation Center for Sustainable Forestry in Southern China, College of Forestry and Grassland College of Soil and Water Conservation, Nanjing Forestry University, Nanjing 210037, China; lfd@njfu.edu.cn (F.L.); szl@njfu.edu.cn (Z.S.); zth@njfu.edu.cn (T.Z.); gaoxiaoge@njfu.edu.cn (X.G.); 2Jiangxi Qiyunshan Food Co., Ltd., Ganzhou 341300, China; 3Forestry Bureau of Chongyi County Jiangxi Province, Ganzhou 341300, China

**Keywords:** *Choerospondias axillaris*, genome size, flow cytometry, K-mer

## Abstract

For the *Choerospondias axillaris* (Roxb.) B.L.Burtt & A.W.Hill, a significant economic tree in the Anacardiaceae family with industrial, medicinal, and ecological value, the genome size remains unreported. Here, we optimized the flow cytometry-based method for ploidy analysis, finding that WPB lysis solution proved to be the most effective. Analysis of 58 *C. axillaris* accessions identified 47 diploids and 11 triploids. The average genome size of diploids was estimated at 450.36 Mb. Illumina sequencing of a diploid (No.22) generated 81.98 Gb of high-quality data (224.44X depth). K-mer analysis estimated the genome size at 365.25 Mb, with 0.91% genome heterozygosity, 34.17% GC content, and 47.74% repeated sequences, indicating high heterozygosity and duplication levels in the genome. Genome assembly may necessitate a combination of second- and third-generation sequencing technologies. Comparative analysis with the NT database revealed that *C. axillaris* exhibited the highest similarity to *C. axillaris* (3.01%) and *Pistacia vera* (2.5%). This study establishes a crucial theoretical framework for *C. axillaris* genome sequencing and molecular genetics.

## 1. Introduction

*C. axillaris* is a significant economic tree species in China, valued for its fruit and wood. As a deciduous tree of the Choerospondias genus (Anacardiaceae family, Sapindales order) [1], it is primarily distributed in southern China (Jiangxi, Fujian, Hunan, Guangxi, and Yunnan provinces) and other Asian countries, including Nepal, Bhutan, Vietnam, Cambodia, and Japan [2]. The majority of *C. axillaris* fruits are oval, turning pale yellow when ripe, with a sweet and sour taste, abundant in nutrients, and possessing both edible and medicinal properties [3]. Rich in nutrients and bioactive compounds such as polyphenols, organic acids, polysaccharides and other chemical constituents [4,5,6,7,8,9], it is used in food products like sour jujube cake, jujube slices, and jelly, while also exhibiting pharmacological properties, including antioxidant, anti-myocardial fibrosis, and anti-tumor effects [3,10]. The bark of the *C. axillaris* tree serves as a raw material for leather tanning and tannin extraction, while the wood exhibits a straight grain and soft texture [11]. Despite these valuable attributes, research on *C. axillaris* remains insufficient, particularly at the genomic and molecular levels.

Methods for identifying chromosome ploidy include morphological, cytological, and molecular approaches [12,13,14,15]. Chromosome count is the most direct and accurate method, but the process is complicated, the technical proficiency is higher, and the time is longer [16]. Recently, flow cytometry using fluorescent-labeled nuclear suspensions has gained popularity for ploidy identification [17]. This method is less time-consuming and technically demanding, with a straightforward preparation process, making it a highly efficient ploidy detection method [18,19,20]. Successful flow cytometry analysis hinges on the quality of the nuclei suspension, influenced by the lysate and preparation technique, which can vary among species [21,22]. Currently, there is a lack of systematic investigation into the flow cytometry method on *C. axillaris*.

Genomic research on *C. axillaris* has primarily focused on the structural and genetic composition analysis of its chloroplast genome [23], with limited comprehensive analysis of its nuclear genome. Given the resource-intensive nature of whole-genome sequencing for *C. axillaris*, developing a tailored sequencing strategy based on genome size and complexity of *C. axillaris* is essential [24,25]. Common methods for genome size estimation include flow cytometry and K-mer analysis of genome survey sequencing [26,27]. Flow cytometry is widely preferred due to its simplicity, speed, accuracy, and cost-effectiveness, and is extensively applied in plant genome size assessment [21]. However, selecting appropriate internal references and considering factors such as test materials, lysate types, and treatment conditions are crucial when employing this method across diverse plant genomes [26,27]. The method based on K-mer analysis, derived from high-throughput sequencing and statistical principles, offers high reliability but may be prone to errors stemming from analysis software [28,29]. Therefore, integrating both methods enables precise determination of key plant genome parameters, encompassing total genome size, GC content, heterozygosity level, and relative abundance of repeated sequences [30,31]. These insights not only underpin genome sequencing endeavors but also establish a robust theoretical framework for plant molecular genetics investigations.

In this study, we optimized flow cytometry for *C. axillaris*, including the determination of materials, the type of nuclear lysate and the dissociation time, and the optimized method was used to identify 58 germplasm resources. The genome size of diploid *C. axillaris* was determined for the first time using fluorescence intensity estimation in conjunction with flow cytometry and K-mer analysis. Additionally, key genomic features such as heterozygosity, GC content, and repeat sequence ratio of *C. axillaris* were assessed through detailed analysis of Illumina high-throughput sequencing data. BLAST alignment was used to assess sequence similarity with other plants. The results provide critical data reference for formulating the whole-genome sequencing strategies and constructing the fine-scale genome map, advancing molecular biology research on *C. axillaris*.

## 2. Results

### 2.1. Establishment of Flow Cytometry Analysis Method

#### 2.1.1. Determination of Test Materials

The process of dissociation eliminates interfering chemicals from cells and disperses them after physical and chemical treatment. Three stages of *C. axillaris* leaves were collected for this experiment: the leaf emergence period (slight curl of leaves, obvious redness can be observed), leaf expansion molding (expansion of leaves, tender green color), and fully expanded leaves (wide upper and narrow lower, dark green color), and the flower organs of *C. axillaris* were used to evaluate the detection conditions (Figure 1, Appendix A). The findings demonstrated that the leaves of the leaf expansion stage could be used as detection materials (Figure 2A–C), while the floral organs of *C. axillaris* were not suitable for flow cytometry (Figure 2D, Appendix A).

#### 2.1.2. Screening of Dissociation Solution

The three dissociation solutions of mGb, WPB, and LB01 (Leagene Biotechnology Co., Ltd., Beijing, China) were chosen for the test after the leaves of *C. axillaris* were formed. The cell number and dispersion of mGb and WPB dissociation solution were appropriate (Figure 3A,B), suggesting that both of them could be used to dissociate *C. axillaris* leaves. However, the number of cells dissociated from the LB01 dissociation solution was small, making it unsuitable for the dissociation of *C. axillaris* leaves (Figure 3C). Further comparison of the dissociation effect of mGb and WPB (Figure 3D,E) revealed that the absorption peak of WPB was narrow and sharp, the impurity peak was small, and the effect was better (Figure 3E, Appendix A).

#### 2.1.3. Determination of Dissociation Time

The leaves were rapidly cut with a blade, the dissociation solution was added, and the dye was applied right away. After testing the machine (designated as dissociation 0 min), the dye was injected, and the static dissociation times were 5 and 10 min. To determine the proper dissociation time of *C. axillaris* leaves, the dyeing time was measured using the orthogonal test. Direct dyeing has the best dissociation impact on the machine 0 min after chopping (Figure 4A,B). The miscellaneous peaks and cell apoptosis both increased significantly after 5 min of separation (Figure 4C,D, Appendix A). Therefore, the experimental operation should be completed as soon as feasible after cutting, and the experimental effect can be guaranteed without delay.

### 2.2. Ploidy Detection and Genome Size Estimation of C. axillaris

The rice (*Oryza sativa* subsp. japonica ‘Nipponbare’, DNA 2C = 0.91 pg) and tomato (*Solanum lycopersicum* L. LA2683, DNA 2C = 0.92 pg) with known genome size were used as internal standard materials. Based on the optimized flow cytometry analysis, 47 of the 58 *C. axillaris* germplasm resources were identified as diploid and 11 as triploid using rice internal reference standards. To confirm the ploidy of these 11 putative triploids, we used tomato as standard, yielding fully consistent results and thus validating the initial findings. The genome size and ploidy of 58 *C. axillaris* germplasm resources were calculated (Appendix A). The coefficient of variation value of flow cytometry analysis ranged from 2.4% to 6.9%, indicating that the experimental results were consistent and dependable. Among them, the ploidy coefficient of 47 samples was 0.91~1.15, and the ploidy estimation value was 1.81~2.29, which was judged as diploid, accounting for 81.03% (Figure 5A and Appendix A). The genome size of diploid was between 402.85 and 563.89 Mb (Appendix A), with an average of 450.36 ± 28.51 Mb (Appendix A). The ploidy coefficient of eleven samples was 1.27~1.66, and the estimated ploidy value was 2.53~3.32, which was judged to be triploid, accounting for 18.97% (Figure 5B and Appendix A). The genome size of triploid was between 571.01 and 746.64 Mb (Appendix A), with an average of 678.51 ± 61.41 Mb.

### 2.3. Survey Analysis of Genome Size of C. axillaris

#### 2.3.1. Sequencing Quality and GC Content

The genome of *C. axillaris* (No.22) was sequenced by high-throughput sequencing platform, and 82.55 Gb raw data was obtained. A total of 81.98 Gb effective data was obtained after Fastp filtering. The proportion of sequencing data Q20 was more than 98%, and the proportion of Q30 was more than 95% (Table 1, indicating that the genome sequencing data was good. The quality of the filtered sequencing data was assessed by FastQC, and the quality values of most sequencing data from *C. axillaris* genome were >34 (Appendix A). The detection of base content distribution showed that except for the large fluctuation of the first few bases (caused by random primer amplification deviation), the proportion of subsequent bases A and T basically coincided, and the proportion of G and C also basically coincided (Figure 6A,B), indicating that there was no AT and GC separation phenomenon in the paired-end sequencing sequence, and the sequencing bases met the requirements. The GC content was 34.17%, which is moderate and will not lead to sequence offset (Figure 6C,D). HiFi sequencing data demonstrates a strong lack of linear correlation between GC content and sequencing depth (Figure 6C). The data points were tightly clustered, forming a horizontal band, which indicates that sequencing depth remains stable across the entire GC content range (20–70%). No significant decrease or increase in depth was observed due to extremely high or low GC content. These results suggest a successful HiFi sequencing experiment with high-quality library construction, minimal GC bias, and uniform and comprehensive coverage of the genome. Analysis of the GC-sequencing depth distribution from the NGS data revealed a bimodal pattern (Figure 6D), a known artifact associated with PCR amplification bias during Illumina library construction. This bias leads to reduced coverage in genomic regions with extreme GC content. To confirm the technical nature of this observation, we compared it to HiFi long-read data from the same sample, which, being generated from a PCR-free library, exhibited a uniform unimodal distribution (Figure 6C). Despite this bias, coverage was sufficient across the genome, and all downstream analyses were based on well-covered regions, minimizing any potential impact on variant calling.

#### 2.3.2. Sample Contamination Assessment

A total of 50,000 sequences were randomly selected from the valid data for homologous alignment in the NT database. The top five species were all from Anacardiaceae family species except *Ailanthus altissimus* (Table 2). The species with the highest sequence alignment rate was the chloroplast genome of *C. axillaris* (3.01%), followed by *Pistacia vera* (2.5%), indicating that *Pistacia vera* was closely related to *C. axillaris*. The other species in the comparison results were all plants, indicating that the sample sequencing data was trustworthy for the K-mer analysis that followed and was not tainted.

#### 2.3.3. Assessment of Genome Size and Heterozygosity

Using 81.98 Gb clean data, K = 19 was used for analysis. After Jellyfish analyzed the sequencing reads, the total number of K-mer was 68,365,809,177 (Table 3). GenomeScope2 was used to visualize the K-mer frequency distribution (Figure 7A). The horizontal axis represents the depth of K-mers, while the vertical axis indicates the frequency of K-mers. The K-mer distribution curve for *C. axillaris* date exhibits deviation from a Poisson distribution, displaying two prominent peaks at depths of approximately 96.8 and 183, respectively. According to the total number of K-mers and the depth of the main peak K-mer at 183, the genome size of *C. axillaris* was preliminarily estimated to be approximately 365.25 Mb. The sequencing depth was estimated to be about 224.44× based on the estimated genome size. GenomeScope2 further estimated the genome heterozygosity and repeat sequence content of *C. axillaris*. The heterozygous ratio of *C. axillaris* was 0.91%, and the duplication ratio was 47.74% (Table 3). The K-mer frequency distribution curve revealed a significant heterozygous peak at 1/2 the expected depth (Figure 7A), suggesting increased genetic complexity in the genome of *C. axillaris*. The tailing effect observed beyond the main peak indicates the presence of repetitive sequences. These findings suggest that the genome of *C. axillaris* is characterized by high heterozygosity and a high content of repetitive sequences. Moreover, ploidy level was assessed using Illumina sequencing reads through the Smudgeplot method, which estimates ploidy based on the ratio of heterozygous k-mer pairs. Analysis with a k-mer size of 19, focusing on the most abundant k-mer pairs, indicated that the genome of *C. axillaris* is in a heterozygous diploid (AB) form (Figure 7B), consistent with the flow cytometry results.

## 3. Discussion

### 3.1. Establishment of Flow Cytometry Method for C. axillaris

The tissue structure and chemical composition of plants vary significantly, necessitating the selection of an appropriate lysis solution to achieve an ideal cell suspension [32]. Past research shows that WPB was optimal for flow cytometry analysis in some contexts. For example, WPB performed better in recalcitrant tissues from woody plants [33]. And in *Bougainvillea* Comm. ex Juss., WPB lysis buffer successfully isolated more intact nuclei, making it the optimal choice for flow cytometry analysis in this species [19]. LB01 buffer was determined to be the most accurate for four bryophyte species (*Brachythecium velutinum*, *Fissidens taxifolius*, *Hedwigia ciliata*, and *Thuidium minutulum*) [34]. Furthermore, LB01 and Otto’s buffers were reported to be the best for young leaf tissue of *Sedum burrito*, *Lycopersicon esculentum*, *Celtis australis*, *Pisum sativum*, *Festuca rothmaleri*, and *Vicia faba* [35]. However, in mature grape leaves, nuclei populations could not be distinguished when using LB01 buffer due to metabolite interference [36]. However, there is no universally applicable lysis solution that currently exists. Notably, sodium citrate has been shown to eliminate RNA interference in suspensions, which was only present in mGb and WPB lysates (Table 4). In this study, the quality of the peak images obtained from both lysates was relatively high, indicating the suitability of these components for *C. axillaris* leaves. Triton X-100, a detergent capable of extracting cell membrane proteins, effectively releases nuclei, prevents nuclear adhesion, and maintains nuclear integrity [37]. Previous studies have indicated that the concentration of Triton X-100 in LB01 lysate significantly enhances nuclear release in Kadsura plants [38]. Notably, the WPB lysate contained a higher concentration of Triton X-100 (Table 4) compared to the mGb lysate, resulting in a greater dissociation of nuclei from *C. axillaris* samples in this study. Based on comprehensive comparison and analysis, the WPB lysis solution is identified as the most appropriate choice for *C. axillaris* leaves.

Determining the ploidy of *C. axillaris* is fundamental for further research on its reproductive development characteristics, as well as for hybridization breeding or ploidy breeding. In our study, ploidy identification was performed for the first time on 58 accessions of *C. axillaris* germplasm resources. The results showed that the coefficient of variation (CV) values measured by flow cytometry ranged from 2.4% to 6.9%. Previous studies have indicated that flow cytometry results remain reliable when CV values are within 9.0% [39], demonstrating that the pretreatment and detection methods established in this experiment are robust. Based on genome size estimation, 11 samples exhibited a ploidy index between 1.27 and 1.66, with an estimated ploidy level ranging from 2.53 to 3.32, indicating they are triploid. Meanwhile, 47 samples showed a ploidy index between 0.91 and 1.15, with an estimated ploidy level of 1.81 to 2.29, identifying them as diploid. Although no polyploid reports of *C. axillaris* were found in earlier studies, natural triploids have been documented in other Anacardiaceae species, such as *T. vernicifluum* [40]. These findings suggest that the *C. axillaris* germplasm resources assessed in this study exhibit rich genetic diversity based on flow cytometry identification, which can provide diverse parental materials for breeding new varieties.

### 3.2. Flow Cytometry and K-Mer Analysis Estimate the Genome Size

High-quality genome sequencing has advanced research in plant molecular biology. Understanding genome size is a prerequisite for genome sequencing. Flow cytometry is a commonly used method for estimating genome size [41], but the results are influenced by various factors. We first optimized the flow cytometry protocol for *C. axillaris*, and the results showed that using WPB lysis buffer on tender leaves with a lysis time of 5 min yielded the best outcomes. This is consistent with findings reported in studies on *Celtis australis* [33] and *Bougainvillea Comm*. ex Juss. [34]. Using the optimized flow cytometry method, we conducted ploidy analysis on 58 accessions of *C. axillaris*, identifying 47 diploids and 11 triploids. Natural triploids primarily arise from the formation of unreduced gametes due to meiotic failure, followed by fertilization with normal gametes [42]. Alternatively, they may result from interspecific distant hybridization leading to the production of unreduced gametes [43]. However, further morphological and genetic evidence is needed to fully elucidate the mechanisms underlying polyploid formation in *C. axillaris*. Currently, four Anacardiaceae species have been sequenced and assembled at the chromosome level: *Toxicodendron vernicifluum* (491 Mb) [44], *Rhus chinensis* (389.40 Mb) [45], *Mangifera indica* (374.8–396 Mb) [46,47,48], and *Pistacia vera* (596–671 Mb) [49,50]. Additionally, draft genome assemblies have been completed for three other Anacardiaceae members: *Sclerocarya birrea* (330.9 Mb) [51], *Anacardium occidentale* (356.6 Mb) [52], *Toxicodendron radicans* (391.3 Mb) [53], and *Mangifera persiciforma* (382.84 M) [54]. These data reveal that genome sizes within the Anacardiaceae family range from 330.9 to 671 Mb, demonstrating that *C. axillaris* exhibits genome size characteristics typical of this plant family.

With the advancement of sequencing technologies, genome survey analysis (k-mer) has further provided estimates of genome size. The sequencing yielded a total of 110.6 Gb of HiFi clean data and 81.98 Gb of NGS clean data, with depths of 302.79× and 224.44×, respectively, indicating high-quality genome sequencing. Alignment of 50,000 randomly selected reads from *C. axillaris* against the NCBI database revealed that the top five matching species belonged to the Anacardiaceae family, with the exception of *Ailanthus altissimus* (Table 2). The highest alignment rate was observed with the *C. axillaris* chloroplast genome (3.01%) [23], followed by *Pistacia vera* (2.5%), suggesting a close phylogenetic relationship between these species. The exclusive identification of plant species in the comparison results confirmed the absence of contamination in the sequencing data, validating its reliability for subsequent K-mer analysis. Furthermore, while *C. axillaris* exhibited the highest read matching rates with *Pistacia vera* (2.5%) and *Ailanthus latissimus* (1.32%), the overall similarity did not exceed 5%. This limited similarity likely reflects the scarcity of sequence information for *C. axillaris* and its related species in the NCBI database, highlighting the need for expanded genomic data to enhance alignment accuracy and coverage. It is noteworthy that the K-mer estimates are slightly lower than those obtained through flow cytometry. This discrepancy may be attributed to the influence of secondary metabolites in *C. axillaris* or the choice of internal reference species. Similar observations have been reported in genome size studies of cucumber (*Cucumis sativus* L.) and other species, where flow cytometry values consistently exceeded those derived from K-mer analysis [28,55]. Nevertheless, the genome size estimates from both methods are closely aligned, providing a reliable foundation for subsequent genome assembly and comparative analysis [56]. The heterozygosity and the duplication ratio observed in *C. axillaris* provide critical insights into its genomic complexity. Our results revealed the heterozygosity level of 0.91%, which is higher than that reported for other economically important Anacardiaceae species, such as *Toxicodendron vernicifluum* (0.56%) [46] and *Rhus chinensis* Mill (0.83%) [47], while lower than Mango (1.5%) [48], *Pistacia vera* (1.72%) [49], and *Mangifera persiciforma* (2.35%) [54]. This suggests that *C. axillaris* may exhibit a capacity for genetic adaptation, potentially influencing its response to environmental stresses or domestication efforts. Similarly, the duplication ratio (47.74%) aligns with trends observed in Mango (47.28%) [50], *Anacardium occidentale* (46%) [52], and *Sclerocarya birrea* (45.18%) [51], but diverges from the higher repeat abundance in *Toxicodendron vernicifluum* (61.66%) [44] and *Rhus chinensis* (56.13%) [45]. These differences may reflect evolutionary adaptations to specific ecological niches or reproductive strategies within the Anacardiaceae family. The high heterozygosity of *C. axillaris* implies significant genetic diversity within natural populations, which could be harnessed for trait improvement, such as fruit yield and disease resistance. Marker-assisted selection (MAS) targeting heterozygous loci may accelerate breeding cycles, while genome-wide association studies (GWAS) could identify alleles linked to medicinal or nutritional properties [57]. The repeat content and heterozygosity patterns suggest that *C. axillaris* populations may harbor unique genetic reservoirs vulnerable to habitat fragmentation. Prioritizing in situ conservation of high-diversity populations would mitigate genetic erosion [58].

K-mer analysis revealed that the *C. axillaris* genome exhibits characteristics of a complex genome with both high repetitive content and heterozygosity. For such genomes, we recommend employing third-generation PacBio Sequel sequencing technology coupled with chromosome conformation capture (Hi-C) to achieve chromosome-level assembly. The genome size estimation of *C. axillaris* not only establishes a foundation for genome sequencing and assembly, but also provides critical data for subsequent studies on genetic evolution, polyploidization events, and molecular mechanisms underlying its distinctive phenotypic traits.

## 4. Materials and Methods

### 4.1. Sample Collection

The germplasm resources of *C. axillaris* were collected from the National Germplasm Resources Bank of *C. axillaris* in Chongyi County, Ganzhou City, Jiangxi Province (25°41′ N, 114°18′ E). The leaves at different stages and floral organs of No.22 (QYS13) diploid plants collected from the *C. axillaris* National Germplasm Resources Bank were used to optimize the flow cytometry identification method of *C. axillaris* (Appendix A). In May 2023, a total of 58 fresh tender leaves were collected (Appendix A), put in silica gel, and immediately frozen No.22, transported back to Nanjing Forestry University by cold chain express delivery, and subsequently stored under conventional conditions for flow cytometry analysis. The rice (*Oryza sativa* subsp. japonica ‘Nipponbare’, DNA 2C = 0.91 pg) and tomato (*Solanum lycopersicum* L. LA2683, DNA 2C = 0.92 pg) were used as internal standard materials.

### 4.2. Establishment of Flow Cytometry Method

#### 4.2.1. Sample Preparation and Testing

Take 0.5–1.0 cm^2^ leaves of an internal standard and the sample to be tested, respectively, and place them in an ice-cold Petri dish. Add 1 mL of pre-chilled lysis solution, and promptly mince the leaves vertically using a sharp double-sided blade, ensuring immersion in the lysis solution. Pipette the homogenized solution and pass it through a 0.04 mm pore size filter to obtain a cell nucleus suspension. Subsequently, transfer the suspension to an assay tube, add 20 μL of propidium iodide (PI) dye, thoroughly mix, and incubate in the dark at 4 °C for one minute. Following incubation, analyze the sample using a Canto II flow cytometer (BD, Franklin Lakes, NJ, USA). Excite the stained samples with 488 nm wavelength light, capture fluorescence in the PE channel (585/42), measure PI emission fluorescence intensity, and analyze 5000–10,000 cells per detection.

#### 4.2.2. Screening of Detection Materials

Leaves and flower organs at various developmental stages of *C. axillaris* were chosen as experimental materials (Figure 1). Each treatment was replicated three times to identify the optimal detection sites for flow detection of *C. axillaris* (Appendix A). The experimental protocol followed the procedures outlined in Section 4.2.1.

#### 4.2.3. Screening of Detection Dissociating Liquids

In this investigation, lysates mGb, LB01 and WPB (Beijing Reagan Shopping Technology Co., Ltd., Beijing, China) were selected. The composition of each lysate is detailed in Table 4. Each experimental condition was replicated thrice to assess the impact of various lysates on detection efficacy (Appendix A).

#### 4.2.4. Optimizing the Dissociation Time

Following the protocol outlined in Section 4.2.1, three dissociation times were established: 0–5 min and 5–10 min, respectively. The impact of varying dissociation times on detection efficacy was compared, with each treatment repeated three times (Appendix A).

#### 4.2.5. Detection of Genome Size and Ploidy of *C. axillaris* by Flow Cytometry

The optimized flow cytometry analysis method was employed to assess the leaf samples of 58 strains of *C. axillaris*. By analyzing the relative fluorescence intensity peaks of the samples and the genome size of internal standard samples, the genome size of *C. axillaris* was determined. Samples exhibiting peak overlap were reanalyzed using a sequential detection protocol. Specifically, both the target samples and the internal standards were independently processed in separate tubes following identical protocols for preparation, propidium iodide (PI) staining, and lysis. Data acquisition was performed sequentially under consistent instrument settings. Genome size and ploidy were subsequently calculated using non-overlapping peak data derived from these independent measurements. The calculation method involved is as the Formula (1). The ploidy of the sample under examination can be calculated as the Formula (2).DNA content of the sample under examination = DNA content of the internal reference × (fluorescence intensity of the sample under examination/fluorescence intensity of the internal reference sample)(1)Ploidy of the sample under examination = ploidy of a known variety × (DNA content of the sample under examination/DNA content of the known variety(2)

### 4.3. Analysis of C. axillaris Genome Survey

#### 4.3.1. Genomic DNA Extraction

Genomic DNA was isolated from diploid *C. axillaris* leaf (No.22) using the CATB method [32]. The purity of the DNA was assessed by a spectrophotometer (Nanodrop 2000, Thermo Fisher Scientific, Waltham, MA, USA), while its integrity was evaluated with a bioanalyzer (Agilent 2100, Agilent Technologies, Santa Clara, CA, USA). Furthermore, the DNA concentration and total yield were determined using a Qubit fluorescence analyzer (Qubit 4.0, Thermo Fisher Scientific, Waltham, MA, USA), and the distribution of DNA fragment sizes was analyzed through agarose gel electrophoresis.

#### 4.3.2. Sample Sequencing, Data Filtering and Quality Control

The qualified DNA samples were sequenced, and a library was constructed by Wuhan Benagen Technology Co., Ltd. (Wuhan, China). The DNA underwent random fragmentation via ultrasonic disruption to generate a 150 bp insert library. Subsequently, high-throughput paired-end sequencing was conducted using the Huada DNBSEQ-T7 sequencer. To ensure data quality, Fastp (0.20.1) software was employed to eliminate low-quality and linker sequences from the initial sequencing output. Additionally, FastQC (v.11.9) software was utilized for quality assessment of the sequencing data, with the resulting clean data being utilized for subsequent analyses.

The high-quality and purified genomic DNA samples were obtained, and a SART cell sequencing library containing about 15 kb preparation solutions (Pacific Biosciences, Santa Clara, CA, USA) cut fragment was constructed. The library preparation involved the following key steps: (1) fragmentation of genomic DNA; (2) repair of DNA damage, end repair, and A-tailing; (3) adapter ligation using the SMRTbell Express Template Prep Kit 2.0 (Pacific Biosciences); (4) nuclease treatment of the SMRTbell library with the SMRTbell Enzyme Cleanup Kit (Pacific Biosciences, Menlo Park, CA, USA); and (5) size selection and polymerase binding. Sequencing was performed on the PacBio Sequel II platform with Sequencing Primer V2 (New England Biolabs, Ipswic, CA, USA) and the Sequel II Binding Kit 2.0 (Pacific Biosciences, Menlo Park, CA, USA) at the Genome Center of Grandomics. Sequencing data can be found in the National Genomics Data Center under accession number PRJCA031736. A total of 110.6 Gb of HiFi reads were generated and subjected to quality control statistics using SMRTlink, resulting in high-quality, valid data. A GC-depth distribution plot was subsequently generated based on these data. The above sequencing was performed at the Wuhan Benagen Technology Co., Ltd.

#### 4.3.3. K-Mer Analysis and Genome Feature Estimation

The genome size, heterozygosity, and repeat sequence ratio of *C. axillaris* were estimated through K-mer analysis using a K-mer of 19, where 19-base sequences were iteratively extracted from the sequencing data via a sliding window. The estimation of genome size, heterozygosity, and repeat sequence content is achieved by scrutinizing the distribution of K-mer frequencies [31]. The Jellyfish software (v 2.3.0) was employed to analyze the valid data for obtaining the depth distribution of K-mer frequencies. The genome size of the species was calculated using the Formula (3). Subsequently, GenomeScope2 software was utilized to model the K-mer frequency distribution data, enabling the plotting of the K-mer frequency distribution curve to derive additional genome characteristics such as heterozygosity and repeat sequence proportion.genome size = total number of K-mers/average K-mer depth(3)

## 5. Conclusions

In this study, we optimized the flow cytometry protocol for *C. axillaris*, identifying leaves as the optimal detection material and the WPB lysis buffer as the most effective solution. Using this optimized method, the average diploid genome size of *C. axillaris* was determined to be 450.36 Mb, while genome survey analysis yielded a size of 365.25 Mb. K-mer analysis further characterized the genome, revealing the heterozygosity ratio of 0.91% and the duplication ratio of 47.74%, classifying it as a highly heterozygous and repetitive genome. These findings provide valuable insights for developing molecular markers and guiding breeding and improvement strategies for *C. axillaris*. Further research should also focus on the analysis of the genome structure of *C. axillaris*, the discovery of functional genes, and the exploration of genes related to key agronomic traits, with the aim of realizing the efficient use of *C. axillaris* resources and genetic improvement.

## Figures and Tables

**Figure 1 plants-14-03094-f001:**
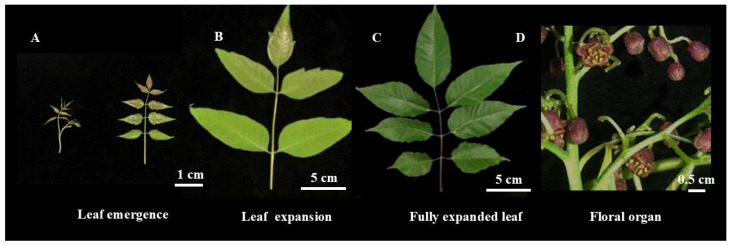
Different development periods’ leaves and flower dissociation materials of *C. axillaris*: (**A**) leaf emergence; (**B**) leaf expansion; (**C**) fully expanded leaf; (**D**) floral organ.

**Figure 2 plants-14-03094-f002:**
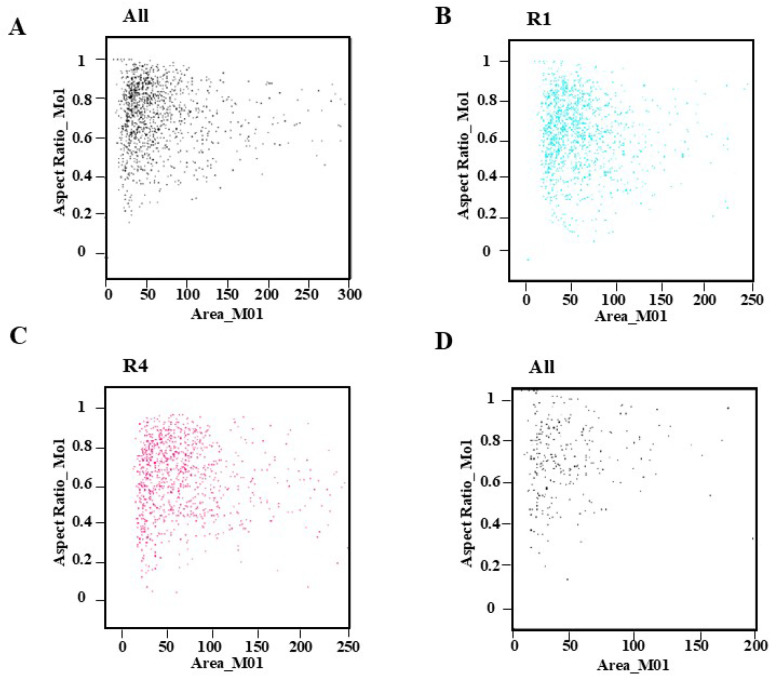
Dot plot of *C. axillaris* leaves and floral organ dissociation: (**A**) leaf emergence; (**B**) leaf expansion; (**C**) fully expanded leaf; (**D**) floral organ. The *X*-axis represents the pulse area of each detected nucleus. The *Y*-axis represents the shape of the fluorescence pulse, calculated as the ratio of pulse width to pulse height.

**Figure 3 plants-14-03094-f003:**
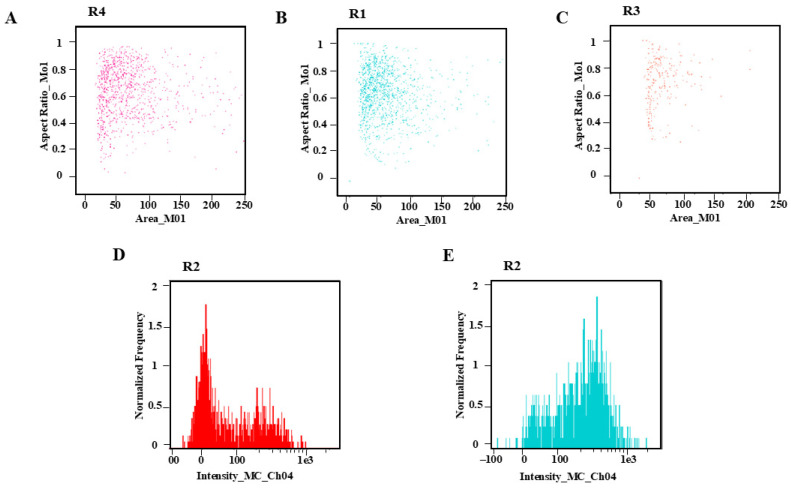
Dot plot and histogram comparison of dissociative fluid species: (**A**) dot plot of mGb; (**B**) dot plot of WPB; (**C**) dot plot of LB01; (**D**) histogram comparison of mGb; (**E**) histogram comparison of WPB. In (**D**,**E**), the *X*-axis represents the fluorescence intensity detected in Channel 4 (Ch04) of the flow cytometer, typically corresponding to a specific fluorescent dye. The “MC” stands for “Measured Channel”, indicating the signal strength proportional to the target molecule. And the *Y*-axis represents the relative frequency of cells scaled to the highest peak. This normalization allows better comparison between different samples by removing absolute count differences.

**Figure 4 plants-14-03094-f004:**
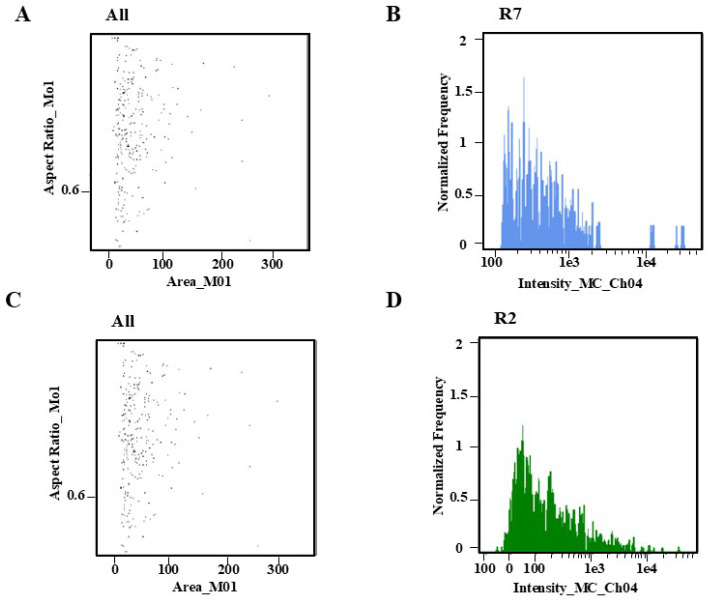
(**A**) Dot plot of 0–5 min; (**B**) histogram comparison of 0–5 min; (**C**) dot plot of 5–10 min; (**D**) histogram comparison of 5–10 min.

**Figure 5 plants-14-03094-f005:**
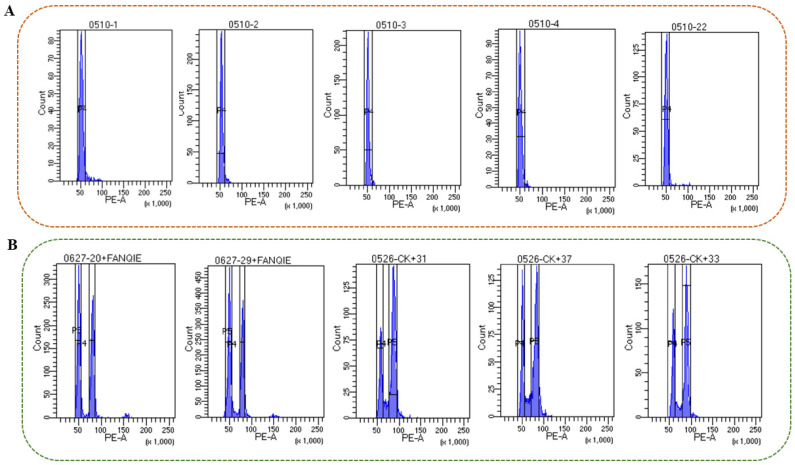
Histogram of 58 *C. axillaris* samples by flow cytometry. (**A**) Five histograms in 47 diploids *C. axillaris* samples; (**B**) Five histograms in eleven triploids *C. axillaris* by flow cytometry. The *X*-axis represents the total fluorescence intensity of each detected nucleus. Higher values indicate larger genome sizes or higher ploidy levels. The *Y*-axis represents the absolute number of cellular events recorded at each fluorescence intensity level. This is a histogram representation where higher peaks indicate more cells with a specific PE signal intensity.

**Figure 6 plants-14-03094-f006:**
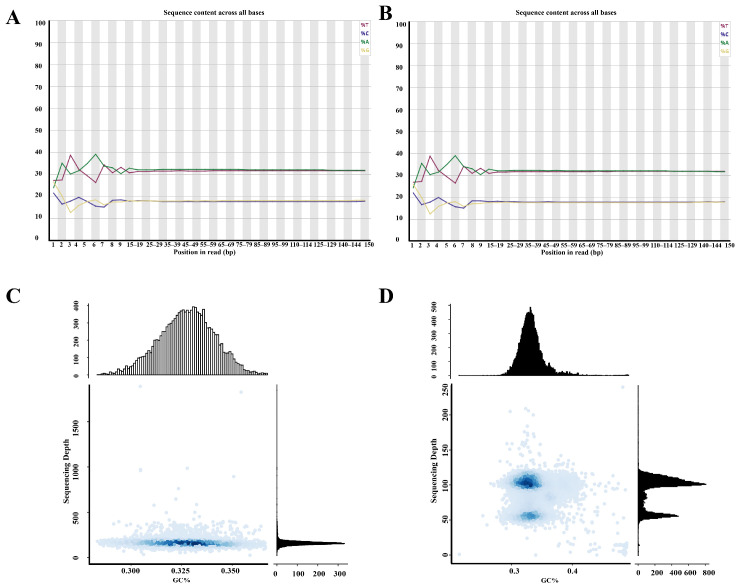
The base content distribution diagram after sequencing data quality control and distribution of GC content. (**A**) The proportion of A, G, C, and T bases at each base position in the first read; (**B**) The proportion of A, G, C, and T bases at each base position in the second read; (**C**) HIFI depth of GC content; (**D**) NGS depth of GC content. In (**A**,**B**), the *X*-axis represents the genomic position in base pairs (bp) along a chromosome or contig. It indicates the physical location where GC content is calculated, typically in sliding windows. And the *Y*-axis represents the percentage of GC bases (Guanine + Cytosine) within each window. Values range from 0% to 100%, reflecting local GC composition. In (**C**,**D**), the *X*-axis represents the percentage of Guanine and Cytosine bases (GC content) in a specific genomic region or sequencing window. And the *Y*-axis represents the average read coverage (number of sequencing reads mapped) to genomic regions with a given GC%.

**Figure 7 plants-14-03094-f007:**
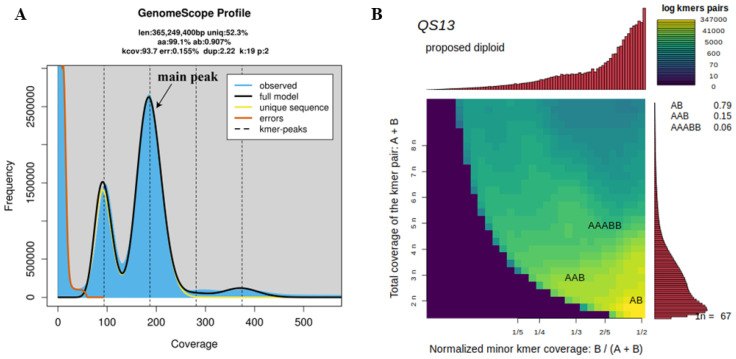
K-mer frequency distribution and ploidy prediction (k = 19). (**A**) K-mer frequency distribution curve when K-mer = 19. The blue area represents the observed k-mer distribution, the black line represents the modeled distribution excluding k-mer errors (shown by the red line), and the yellow line refers to the maximum k-mer coverage stipulated in the model. The *X*-axis represents the sequencing depth of each unique K-mer in the genome assembly. It indicates how many times a particular K-mer is observed in the sequencing data. Higher coverage values typically correspond to repetitive or highly conserved genomic regions. The *Y*-axis shows the number of distinct K-mers observed at each coverage level. This reflects the abundance distribution of K-mers in the dataset. A peak in the histogram often corresponds to the expected coverage of single-copy genomic regions, which can be used to estimate genome size and heterozygosity. (**B**) Two-dimensional heat map was constructed to depict the prediction of ploidy from clean using Smudgeplot. The color intensity corresponds to the approximate amount of k-mers per bin, ranging from purple (weak) to yellow (strong). Estimated ploidies are shown in the upper left corner of each graph, with the likelihood of various ploidies shown in the right.

**Table 1 plants-14-03094-t001:** Statistics of sequencing data of *C. axillaris* genome.

Type	Num_Seqs	Sum_Len/bp	GC_Content (%)	Q20 (%)	Q30 (%)
Raw data	550,344,728	82,551,709,200	34.3	98.43	95.57
Clean data	550,342,818	81,978,145,256	34.17	98.42	95.56

**Table 2 plants-14-03094-t002:** Alignment between the sequencing data of *C. axillaris* and NT database.

Species	Family	Kingdom	Reads	Percentage (%)
*Choerospondias axillaris*	Anacardiaceae	Viridiplantae	1507	3.01
*Pistacia vera*	Anacardiaceae	Viridiplantae	1248	2.5
*Ailanthus altissimus*	Simaroubaceae	Viridiplantae	659	1.32
*Sclerocarya birrea*	Anacardiaceae	Viridiplantae	566	1.13
*Mangifera indica*	Anacardiaceae	Viridiplantae	506	1.01

**Table 3 plants-14-03094-t003:** Statistical data based on K-mer 19 analysis.

Sample	K-Mer	K-Mer Number	Genome Size (M)	Data Size (G)	Heterozygous Ratio (%)	Duplication Ratio (%)	X
No.22	19	68,365,809,177	365.25	81.98	0.91	47.74	224.44

**Table 4 plants-14-03094-t004:** Formulas of dissociating liquids.

Names	Formulas
mGb	45 mmol/L MgCl_2_, 30 mmol/L sodium citrate, 20 mmol/L MPOS, 0.2% (*v*/*v*) Triton X-100, 1% PVP-40, 10 mmol/L Na_2_EDTA·2H_2_O, 20 μL/mL β-mercaptoethanol, pH = 7.0.
LB01	15 mmol/L TRIS, 2 mmol/L Na_2_EDTA, 0.5 mmol/L spermine 4HCL, 80 mmol/L KCl, 20 mmol/L NaCl, 15 mmol/L β-mercaptoethanol, 0.1% (*v*/*v*) Triton X-100, pH = 7.5.
WPB	0.2 mmol/L Tris-HCl, 4 mmol/LMgCl_2_·6H_2_O, 2 mmol/L Na_2_EDTA·2H_2_O, 86 mmol/L NaCl, 10 mmol/L sodium metabisulfite, 1% PVP-10, 1% (*v*/*v*) Triton X-100, pH = 7.5.

## Data Availability

The original contributions presented in the study are included in the article; further inquiries can be directed to the corresponding authors.

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
