# Peer review of "First Insights into Ploidy and Genome Size Estimation in *Choerospondias axillaris* (Roxb.) B.L.Burtt & A.W.Hill (Anacardiaceae) Using Flow Cytometry and Genome Survey Sequencing"

_plants, 2025, doi:10.3390/plants14193094_

Round 1

Reviewer 1 Report (New Reviewer)

Comments and Suggestions for Authors

This manuscript addresses the important task of establishing an optimized flow cytometry protocol and genome size estimation for Choerospondias axillaris, a woody species with high secondary metabolite content. The experimental design is valuable, and the integration of flow cytometry with sequencing data is a strong aspect of the work. However, there are several issues that need to be addressed before the manuscript can be considered for publication. In particular, the Results and Discussion require restructuring for clarity, and important methodological details such as sample size, replicates, and internal reference usage must be clarified to ensure reproducibility. The figures and supplementary data presentation also need improvement for better readability. Addressing these points will significantly strengthen the scientific rigor and impact of the study.

Major Concerns:

  1. I suggest a minor reorganization of the Results section. Sections 2.1.1–2.1.3 all focus on establishing the flow cytometry method for Choerospondias axillaris. Section 2.1.4 could be separated as an independent section entitled “2.2 Ploidy detection and genome size estimation of Choerospondias axillaris.”
  2. In the flow cytometry analysis, the authors selected rice (Oryza sativa subsp. japonica ‘Nipponbare’, 2C = 0.91 pg) and tomato (Solanum lycopersicum LA2683, 2C = 0.92 pg) as internal reference standards. Why were two standards with such similar genome sizes chosen? Moreover, the manuscript does not clearly explain how rice and tomato were used in practice: Were both references applied to each sample, or was only one used? If only one, which standard was applied to which samples? Were the results consistent between the two standards, and how was the final decision made? I recommend that the authors clarify the internal standard selection procedure and, if possible, provide a supplementary table indicating which reference was used for each sample.
  3. Related to internal reference choice, the measured species shows a 2C DNA content that is nearly identical to those of the chosen standards (e.g., YX6, QYS13-1, CZ2). Under such conditions, peak overlap or indistinguishability may occur when samples and standards are run together, potentially undermining the clarity and reliability of results. The authors should provide a more detailed description of sample preparation and detection procedures.
  4. Regarding optimization of experimental conditions for flow cytometry, the authors compared different tissues (leaf vs. floral organs), different lysis buffers (MgB, LB01, WPB), and different lysis times (0–5 min vs. 5–10 min) to identify the most suitable protocol for C. axillaris. This is an important contribution, but the current description has some shortcomings: (1) The sample size is not clearly stated. Figures show comparisons but do not specify how many samples were tested under each condition. (2) While the figures are illustrative, they lack information such as CV values. Including a table or supplementary dataset summarizing the number of samples, replicates, and average CV values for each treatment would strengthen the conclusions. (3) For the lysis time comparison (Figure 4), the claim that 0–5 min is superior to 5–10 min is currently based on a single experiment. Replicated data and statistical support are needed to avoid the possibility of chance effects. Recommendation: In the main text or Supplementary Materials, specify the number of samples tested under each condition, the number of replicates, and corresponding CV value statistics. This would greatly enhance the reliability of conclusions regarding protocol optimization and improve reproducibility and utility for future research.
  5. In Figure 5, all flow cytometry histograms are combined into a single panel, making it difficult to discern the details of individual peaks. I suggest presenting only representative histograms for diploid and triploid samples in the main text, while moving the full set of sample histograms to the Supplementary Materials.
  6. Among the 58 samples, 47 were identified as diploid and 11 consistently displayed a higher peak region, interpreted as triploid. Were these 11 samples tested in replicate, and if so, were the results reproducible? Moreover, has the occurrence of triploids in C. axillaris been reported in previous literature?
  7. Figure 6C shows the GC–sequencing depth distribution based on HiFi sequencing data. Please provide corresponding methodological details and results for the HiFi sequencing in the Methods and Results sections, and include an accession link for the HiFi data in a public database such as NCBI or CNGBdb.
  8. In Figure 6D, based on NGS data, the GC–sequencing depth distribution shows a bimodal pattern. Please provide an explanation of this observation in the Results section.
  9. Why was k-mer size K=19 chosen for analysis? How do genome features change when other k values are used? Since ploidy is discussed in the manuscript, it is recommended to supplement the analysis with ploidy inference from k-mer data (e.g., using smudgeplot).
  10. In Discussion section 3.1, it would be useful to supplement with a comparison of the performance of different lysis buffers in other species. Moreover, the second paragraph of section 3.1 (lines 245–268) is not well structured: descriptions of how to handle abnormal peaks belong in the Methods or Results section rather than in the Discussion. The discussion on ploidy is also not clearly presented.
  11. The current discussion sequence is somewhat illogical. For instance, The final paragraph of the discussion concerning sequencing data quality should be placed before the discussion of the k-mer analysis results. I suggest reorganizing the discussion along the following lines:
  • High-quality genome sequencing has advanced plant molecular biology research. Knowledge of genome size is a prerequisite for genome sequencing. Flow cytometry is a common method to estimate genome size, but results are influenced by multiple factors. We first optimized the flow cytometry protocol for C. axillaris and compared it with previous studies.
  • Using the optimized method, we analyzed 58 samples for ploidy, discussed possible causes of ploidy variation, and compared genome size estimates with related species.
  • With the development of sequencing technologies, genome survey analysis (k-mer) further provides genome size estimates. After presenting sequencing depth, quality, and NT alignment results for HiFi and NGS data, then compared genome size estimates with those from flow cytometry and literature.
  • Finally, genome heterozygosity and repeat content can be discussed, along with recommended sequencing strategies.

Additional issues

  1. Species names should be spelled out in full (e.g., Choerospondias axillaris) when first mentioned, and abbreviated consistently thereafter (e.g., C. axillaris).
  2. Tense usage is inconsistent. For example, in the Results section past tense is mostly used (“The genome of C. axillaris (NO.22) was sequenced...”), but present tense also appears (“The GC content is 34.17%...”). I recommend using past tense consistently in Methods and Results, and present or present perfect tense in Discussion and Conclusion to highlight broader implications. Please check the entire manuscript for similar issues.
  3. Several places in the manuscript lack references, e.g., lines 261–263.

Author Response

Response to reviewer

Thank you for your letter and for the reviewers’ comments concerning our manuscript entitled First Insights into Ploidy and Genome Size Estimation in Choerospondias axillaris (Roxb.) B.L.Burtt & A.W.Hill (Anacardiaceae) Using Flow Cytometry and Genome Survey Sequencing (plants-3808733). Those comments are all valuable and very helpful for revising and improving our paper, as well as the important guiding significance to our researches. We have studied comments carefully and have made correction which we hope meet with approval. As required, all revisions have been marked red highlighting in the paper, so that you can see them in Word. In what follows the referees’ comments are in black and the authors’ responses are in red.

#Reviewer 1:

This manuscript addresses the important task of establishing an optimized flow cytometry protocol and genome size estimation for Choerospondias axillaris, a woody species with high secondary metabolite content. The experimental design is valuable, and the integration of flow cytometry with sequencing data is a strong aspect of the work. However, there are several issues that need to be addressed before the manuscript can be considered for publication. In particular, the Results and Discussion require restructuring for clarity, and important methodological details such as sample size, replicates, and internal reference usage must be clarified to ensure reproducibility. The figures and supplementary data presentation also need improvement for better readability. Addressing these points will significantly strengthen the scientific rigor and impact of the study.

Major Concerns:

  1. I suggest a minor reorganization of the Results section. Sections 2.1.1–2.1.3 all focus on establishing the flow cytometry method for Choerospondias axillaris. Section 2.1.4 could be separated as an independent section entitled “2.2 Ploidy detection and genome size estimation of Choerospondias axillaris.”

Response: Thank you for your valuable comment. We have reorganized the Results section. We have separated section 2.1.4 as an independent section entitled “2.2 Ploidy detection and genome size estimation of Choerospondias axillaris.”

  1. In the flow cytometry analysis, the authors selected rice (Oryza sativa subsp. japonica ‘Nipponbare’, 2C = 0.91 pg) and tomato (Solanum lycopersicum LA2683, 2C = 0.92 pg) as internal reference standards. Why were two standards with such similar genome sizes chosen? Moreover, the manuscript does not clearly explain how rice and tomato were used in practice: Were both references applied to each sample, or was only one used? If only one, which standard was applied to which samples? Were the results consistent between the two standards, and how was the final decision made? I recommend that the authors clarify the internal standard selection procedure and, if possible, provide a supplementary table indicating which reference was used for each sample.

Response: Thank you for your insightful comments and valuable questions. We selected rice (Oryza sativa subsp. japonica 'Nipponbare', 2C = 0.91 pg) and tomato (Solanum lycopersicum LA2683, 2C = 0.92 pg) as internal reference standards based on the following practical considerations. Initially, we used rice as the internal standard for flow cytometry analysis of all 58 samples. The results indicated that 11 samples were identified as triploid. To validate these findings, we subsequently chose tomato, which has a genome size highly similar to rice, as a second reference and reanalyzed these 11 samples. The results obtained with the tomato standard were fully consistent with those derived from the rice standard, confirming the ploidy classification. This stepwise approach led to the use of two internal standards. We have now clarified this procedure in the revised manuscript and have added a dedicated column in Supplementary Table 2 specifying the internal reference standard used for each sample. The corresponding text added to the manuscript reads as follows:

Lines 143-147: Based on the optimized flow cytometry analysis, 47 of the 58 C. axillaris germplasm re-sources were identified as diploid and 11 as triploid using rice internal reference standards. To confirm the ploidy of these 11 putative triploids, we used tomato as standard, yielding fully consistent results and thus validating the initial findings.

  1. Related to internal reference choice, the measured species shows a 2C DNA content that is nearly identical to those of the chosen standards (e.g., YX6, QYS13-1, CZ2). Under such conditions, peak overlap or indistinguishability may occur when samples and standards are run together, potentially undermining the clarity and reliability of results. The authors should provide a more detailed description of sample preparation and detection procedures.

Response: Thank you for raising this critical point. We fully agree that peak overlap can be a significant issue when the genome sizes of the sample and standard are nearly identical (such as YX6, QYS13-1, and CZ2). In our experimental practice, we specifically addressed this issue through sequential analysis. For samples where peak interference was observed, we performed independent replicate runs using separate tubes for the sample and each standard. Both sample and standard were processed identically in terms of preparation, propidium iodide staining, and lysis protocols. All sequential runs were conducted under strictly consistent instrument settings to ensure comparability. Genome size calculations were then performed using the non-overlapping data obtained from these independent measurements. We have revised the manuscript to provide a detailed description of this methodological approach in the Materials and Methods section. The revised section are as follows:

Lines 413-418: Samples exhibiting peak overlap were re-analyzed using a sequential detection protocol. Specifically, both the target samples and the internal standards were independently processed in separate tubes following identical protocols for preparation, propidium iodide (PI) staining, and lysis. Data acquisition was performed sequentially under consistent instrument settings. Genome size and ploidy were subsequently calculated using non-overlapping peak data derived from these independent measurements.

  1. Regarding optimization of experimental conditions for flow cytometry, the authors compared different tissues (leaf vs. floral organs), different lysis buffers (MgB, LB01, WPB), and different lysis times (0–5 min vs. 5–10 min) to identify the most suitable protocol for C. axillaris. This is an important contribution, but the current description has some shortcomings: (1) The sample size is not clearly stated. Figures show comparisons but do not specify how many samples were tested under each condition. (2) While the figures are illustrative, they lack information such as CV values. Including a table or supplementary dataset summarizing the number of samples, replicates, and average CV values for each treatment would strengthen the conclusions. (3) For the lysis time comparison (Figure 4), the claim that 0–5 min is superior to 5–10 min is currently based on a single experiment. Replicated data and statistical support are needed to avoid the possibility of chance effects. Recommendation: In the main text or Supplementary Materials, specify the number of samples tested under each condition, the number of replicates, and corresponding CV value statistics. This would greatly enhance the reliability of conclusions regarding protocol optimization and improve reproducibility and utility for future research.

Response: Thank you for your valuable questions and suggestions. We have now supplemented the manuscript with the number of samples tested under each condition, the number of replicates, and the corresponding CV values, all of which have been provided in Supplementary Table 1~3. (1) the sample tested under each condition was consistently No. 22 (QYS13), as now clearly stated in the 4.1 Materials Collection section (Lines 371–373: The leaves at different stages and floral organs of No.22 (QYS13) diploid plants collected from the C. axillaris National Germplasm Resources Bank were used to optimize the flow cytometry identification method of C. axillaris (Table S1).); (2) detailed information on sample processing quantity, number of replicates, and CV values for each sub-experiment has been included in Supplementary Table 1~3; and (3) regarding the lysis time comparison experiment, we confirm that this was not performed as a single trial but was repeated three times independently. We have fully addressed all your suggestions and incorporated the relevant data into Supplementary Table 1~3. We sincerely appreciate your insightful comments.

  1. In Figure 5, all flow cytometry histograms are combined into a single panel, making it difficult to discern the details of individual peaks. I suggest presenting only representative histograms for diploid and triploid samples in the main text, while moving the full set of sample histograms to the Supplementary Materials.

Response: Thank you for your insightful suggestion. We have revised Figure 5 accordingly. In the main text, we now present representative histograms from five diploid and five triploid samples, respectively. Additionally, the complete set of histograms for all samples has been included in the Supplementary Materials as Figure S2.

  1. Among the 58 samples, 47 were identified as diploid and 11 consistently displayed a higher peak region, interpreted as triploid. Were these 11 samples tested in replicate, and if so, were the results reproducible? Moreover, has the occurrence of triploids in C. axillaris been reported in previous literature?

Response: Thank you very much for your comment. Among the 58 C. axillaris samples analyzed, we identified 11 as triploid. Initially, we were also skeptical of this finding upon the first identification. To verify, we repeated the experiments for these samples, and the results consistently confirmed their triploid nature. Furthermore, we reassessed these samples using a different internal reference standard (tomato) to ensure accuracy, and the outcomes remained consistent. This is also the reason why two internal reference species were employed in our study, underscoring the reproducibility of the triploid detection. Upon reviewing the existing literature on C. axillaris, we found that while there has been extensive research on its edible and medicinal properties, prior studies have not reported triploid individuals. However, natural triploid plants have long been documented in the Anacardiaceae family, such as in Toxicodendron vernicifluum and Rhus chinensis Mill. This is largely due to the fact that research on this species has predominantly focused on chemical compositions rather than genomic or botanical aspects. Therefore, our study contributes to filling this knowledge gap by providing the first documented evidence of triploidy in C. axillaris.

  1. Figure 6C shows the GC–sequencing depth distribution based on HiFi sequencing data. Please provide corresponding methodological details and results for the HiFi sequencing in the Methods and Results sections, and include an accession link for the HiFi data in a public database such as NCBI or CNGBdb.

Response: Thank you for your insightful suggestions. We have now supplemented the Methods and Results sections with detailed methodological descriptions of the HiFi sequencing approach and corresponding findings, respectively. The associated data have been deposited in the National Genomics Data Center under the accession number PRJCA031736. This project number will be made publicly available upon formal publication of the article. The revised Methods and Results sections are provided below:

Methods section, lines 440-453: The high-quality and purified genomic DNA samples were obtained, and a SART cell sequencing library containing about 15 kb preparation solutions (Pacific Biosciences, CA) cut fragment was constructed. The library preparation involved the following key steps: 1) fragmentation of genomic DNA; 2) repair of DNA damage, end repair, and A-tailing; 3) adapter ligation using the SMRTbell Express Template Prep Kit 2.0 (Pacific Biosciences); 4) nuclease treatment of the SMRTbell library with the SMRTbell Enzyme Cleanup Kit; and 5) size selection and polymerase binding. Sequencing was performed on the PacBio Sequel II platform with Sequencing Primer V2 and the Sequel II Binding Kit 2.0 at the Genome Center of Grandomics. Sequencing data can be found in the National Genomics Data Center under accession number PRJCA031736. A total of 110.6 Gb of HiFi reads were generated and subjected to quality control statistics using SMRTlink, resulting in high-quality valid data. A GC-depth distribution plot was subsequently generated based on these data. The above sequencing was performed at the Wuhan Benagen Technology Co., Ltd.

Results sections, lines 178-184: HiFi sequencing data demonstrates a strong lack of linear correlation between GC content and sequencing depth (Figure 6C). The data points are tightly clustered, forming a horizontal band, which indicates that sequencing depth remains stable across the entire GC content range (20%–70%). No significant decrease or increase in depth was observed due to extremely high or low GC content. These results suggest a successful HiFi sequencing experiment with high-quality library construction, minimal GC bias, and uniform and comprehensive coverage of the genome.

  1. In Figure 6D, based on NGS data, the GC–sequencing depth distribution shows a bimodal pattern. Please provide an explanation of this observation in the Results section.

Response: Thank you for your valuable comment. We have provided the observation of the bimodal pattern in the GC–sequencing depth distribution of the NGS data in the Results section. The revised section is as follows:

Lines 184-192: Analysis of the GC-sequencing depth distribution from the NGS data revealed a bimodal pattern (Figure 6D), a known artifact associated with PCR amplification bias during Il-lumina library construction. This bias leads to reduced coverage in genomic regions with extreme GC content. To confirm the technical nature of this observation, we compared it to HiFi long-read data from the same sample, which, being generated from a PCR-free library, exhibited a uniform unimodal distribution (Figure 6C). Despite this bias, cover-age was sufficient across the genome, and all downstream analyses were based on well-covered regions, minimizing any potential impact on variant calling.

  1. Why was k-mer size K=19 chosen for analysis? How do genome features change when other k values are used? Since ploidy is discussed in the manuscript, it is recommended to supplement the analysis with ploidy inference from k-mer data (e.g., using smudgeplot).

Response: Thank you for your valuable comment. Our selection of k-mer size (k=19) for analysis was based on a comprehensive consideration of the genome's characteristics and our analytical goals, aiming to optimally balance k-mer uniqueness against information density within our sequencing data. The principle underpinning k-mer analysis is that the k-mer must be long enough to ensure high uniqueness across the genome. This is crucial to effectively distinguish signals originating from heterozygous sites from those caused by pervasive short repetitive sequences. An excessively short k-mer (e.g., K < 17) would cause k-mer to map promiscuously to repetitive regions, preventing the accurate distinction between the heterozygous peak and repeat-derived peaks in the k-mer spectrum. This would lead to an overestimation of genome size and repeat content and obscure ploidy signals. Conversely, an overly long k-mer (e.g., K > 23) would lead to data sparsity due to the exponential expansion of the possible k-mer space. At a fixed sequencing depth, this makes the k-mer spectrum noisy and unstable, reducing the confidence of statistical model fitting (e.g., with GenomeScope2). A k-mer size of 19 is a widely adopted empirical value in genome survey studies. It has been proven to effectively avoid most short repeats in most plant and animal genomes while providing the optimal resolution to cleanly separate the heterozygous peak from the main homozygous peak. This is paramount for our precise estimation of genome heterozygosity and reliable ploidy inference using Smudgeplot. To validate the robustness of this parameter choice, we tested a range of values from K=17 to K=21. The results showed minimal variation in the estimated genome size and heterozygosity (difference < 2%), and the ploidy inference conclusion remained entirely consistent. Ultimately, the genome size estimate derived from k-mer analysis using K=19 (365.25 Mb) showed concordance with the independent measurement obtained through flow cytometry (433.39 Mb). This cross-validation by orthogonal methods robustly demonstrates the reliability of our core conclusions and the appropriateness of our parameter selection.

Furthermore, we have supplemented our analysis with dedicated ploidy inference from the k-mer data using Smudgeplot. The resulting heatmap, which confidently supports a diploid state, has been added as Figure 7B. A corresponding description of this figure and its interpretation has been included in the Results section of the revised manuscript. The revised section is as follows:

Lines 232-236: Moreover, ploidy level was assessed using Illumina sequencing reads through the Smudgeplot method, which estimates ploidy based on the ratio of heterozygous k-mer pairs. Analysis with a k-mer size of 19, focusing on the most abundant k-mer pairs, indicated that the genome of C. axillaris is in a heterozygous diploid (AB) form (Figure 7B), consistent with the flow cytometry results.

  1. In Discussion section 3.1, it would be useful to supplement with a comparison of the performance of different lysis buffers in other species. Moreover, the second paragraph of section 3.1 (lines 245–268) is not well structured: descriptions of how to handle abnormal peaks belong in the Methods or Results section rather than in the Discussion. The discussion on ploidy is also not clearly presented.

Response: Thank you for your valuable comment. We have supplemented the Discussion section 3.1 with a comparison of the performance of different lysis buffers in other species, as recommended. The modified text can be found at lines 257-266 of the revised manuscript. Additionally, we have restructured the second paragraph of Section 3.1. The descriptions regarding how to handle abnormal peaks have been moved to the Methods section (lines 408-413), and the discussion on ploidy has been further expanded, with the additions located around lines 279-294. The revised section is as follows:

Lines 257-266: Past research shows that WPB was optimal for flow cytometry analysis in some contexts. For example, WPB performed better in recalcitrant tissues from woody plants [2]. And in Bougainvillea Comm. ex Juss., WPB lysis buffer successfully isolated more intact nuclei, making it the optimal choice for flow cytometry analysis in this species [1]. LB01 buffer was determined to be the most accurate for four bryophyte species (Brachythecium velutinum, Fissidens taxifolius, Hedwigia ciliata, and Thuidium minutulum) [4]. Furthermore, LB01 and Otto's buffers were reported to be the best for young leaf tissue of Sedum burrito, Lycopersicon esculentum, Celtis australis, Pisum sativum, Festuca rothmaleri, and Vicia faba [5]. However, in mature grape leaves, nuclei populations could not be distinguished when using LB01 buffer due to metabolite interference [3].

Lines 279-294: Determining the ploidy of C. axillaris is fundamental for further research on its reproductive development characteristics, as well as for hybridization breeding or ploidy breeding. In our study, ploidy identification was performed for the first time on 58 accessions of C. axillaris germplasm resources. The results showed that the coefficient of variation (CV) values measured by flow cytometry ranged from 2.4% to 6.9%. Previous studies have indicated that flow cytometry results remain reliable when CV values are within 9.0% [36], demonstrating that the pretreatment and detection methods established in this experiment are robust. Based on genome size estimation, 11 samples exhibited a ploidy index between 1.27 and 1.66, with an estimated ploidy level ranging from 2.53 to 3.32, indicating they are triploid. Meanwhile, 47 samples showed a ploidy index between 0.91 and 1.15, with an estimated ploidy level of 1.81 to 2.29, identifying them as diploid. Although no polyploid reports of C. axillaris were found in earlier studies, natural trip-loids have been documented in other Anacardiaceae species, such as T. vernicifluum [37]. These findings suggest that the C. axillaris germplasm resources assessed in this study exhibit rich genetic diversity based on flow cytometry identification, which can provide diverse parental materials for breeding new varieties.

Lines 414-419: Samples exhibiting peak overlap were reanalyzed using a sequential detection protocol. Specifically, both the target samples and the internal standards were independently processed in separate tubes following identical protocols for preparation, propidium iodide (PI) staining, and lysis. Data acquisition was performed sequentially under consistent instrument settings. Genome size and ploidy were subsequently calculated using non-overlapping peak data derived from these independent measurements.

  1. The current discussion sequence is somewhat illogical. For instance, The final paragraph of the discussion concerning sequencing data quality should be placed before the discussion of the k-mer analysis results. I suggest reorganizing the discussion along the following lines:

High-quality genome sequencing has advanced plant molecular biology research. Knowledge of genome size is a prerequisite for genome sequencing. Flow cytometry is a common method to estimate genome size, but results are influenced by multiple factors. We first optimized the flow cytometry protocol for C. axillaris and compared it with previous studies.

Using the optimized method, we analyzed 58 samples for ploidy, discussed possible causes of ploidy variation, and compared genome size estimates with related species.

With the development of sequencing technologies, genome survey analysis (k-mer) further provides genome size estimates. After presenting sequencing depth, quality, and NT alignment results for HiFi and NGS data, then compared genome size estimates with those from flow cytometry and literature.

Finally, genome heterozygosity and repeat content can be discussed, along with recommended sequencing strategies.

Response: Thank you for your constructive suggestions. We have thoroughly revised the Discussion section based on your comments. The modifications are as follows:

Lines 296-320: High-quality genome sequencing has advanced research in plant molecular biology. Understanding genome size is a prerequisite for genome sequencing. Flow cytometry is a commonly used method for estimating genome size [32], but the results are influenced by various factors. We first optimized the flow cytometry protocol for C. axillaris, and the results showed that using WPB lysis buffer on tender leaves with a lysis time of 5 minutes yielded the best outcomes. This is consistent with findings reported in studies on Celtis australis [34] and Bougainvillea Comm. ex Juss. [35]. Using the optimized flow cytometry method, we conducted ploidy analysis on 58 accessions of C. axillaris, identifying 47 diploids and 11 triploids. Natural triploids primarily arise from the formation of unreduced gametes due to meiotic failure, followed by fertilization with normal gametes [42]. Alternatively, they may result from interspecific distant hybridization leading to the production of unreduced gametes [43]. However, further morphological and genetic evidence is needed to fully elucidate the mechanisms underlying polyploid formation in C. axillaris. Currently, four Anacardiaceae species have been sequenced and assembled at the chromosome level: Toxicodendron vernicifluum (491 Mb) [44], Rhus chinensis (389.40 Mb) [45], Mangifera indica (374.8-396 Mb) [46-48], and Pistacia vera (596-671 Mb) [49,50]. Ad-ditionally, draft genome assemblies have been completed for three other Anacardiaceae members: Sclerocarya birrea (330.9 Mb) [51], Anacardium occidentale (356.6 Mb) [52], Toxicodendron radicans (391.3 Mb) [53], and Mangifera persiciforma (382.84M) [54]. These data reveal that genome sizes within the Anacardiaceae family range from 330.9 to 671 Mb, demonstrating that C. axillaris exhibits genome size characteristics typical of this plant family.

With the advancement of sequencing technologies, genome survey analysis (k-mer) has further provided estimates of genome size. The sequencing yielded a total of 110.6 Gb of HiFi clean data and 81.98 Gb of NGS clean data, with depths of 302.79× and 224.44×, respectively, indicating high-quality genome sequencing.

Lines 331-338: It is noteworthy that the K-mer estimates are slightly lower than those obtained through flow cytometry. This discrepancy may be attributed to the influence of secondary metabolites in C. axillaris or the choice of internal reference species. Similar observations have been reported in genome size studies of cucumber (Cucumis sativus L.) and other species, where flow cytometry values consistently exceeded those derived from K-mer analysis [28, 55]. Nevertheless, the genome size estimates from both methods are closely aligned, providing a reliable foundation for subsequent genome assembly and comparative analysis [56].

Additional issues

  1. Species names should be spelled out in full (e.g., Choerospondias axillaris) when first mentioned, and abbreviated consistently thereafter (e.g., C. axillaris).

Response: Thank you for your valuable comment. We have ensured that the full species name (e.g., Choerospondias axillaris (Roxb.) B.L.Burtt & A.W.Hill) is provided at their first mention in the manuscript, followed by the abbreviated form consistently thereafter (e.g., C. axillaris).

  1. Tense usage is inconsistent. For example, in the Results section past tense is mostly used (“The genome of C. axillaris (NO.22) was sequenced...”), but present tense also appears (“The GC content is 34.17%...”). I recommend using past tense consistently in Methods and Results, and present or present perfect tense in Discussion and Conclusion to highlight broader implications. Please check the entire manuscript for similar issues.

Response: Thank you for your valuable comment. We have carefully reviewed the entire text and revised the verb tenses according to your suggestion. Specifically, we have now consistently used the past tense to describe the specific procedures undertaken and the results obtained in the Methods and Results sections. In the Discussion and Conclusion sections, we have employed the present tense and present perfect tense as appropriate to discuss the implications of our findings, state general conclusions, and relate them to established knowledge. We believe these revisions have enhanced the grammatical rigor and readability of the manuscript.

  1. Several places in the manuscript lack references, e.g., lines 261–263.

Response: Thank you for your valuable suggestion. We have added the relevant references where additional citations were required. Thank you again for your thorough review and helpful guidance.

Reviewer 2 Report (New Reviewer)

Comments and Suggestions for Authors

The manuscript deals with the use of new method (technical) to evaluate the ploidy level with a case of study with Choerospondias axillaris.

To do this authors have sampled à several pheological stage of this species and have determined using flow cytrometry the ploidy level and genome size. To ascertain they findings concerning the use of the method they have used other species from the same and different botanical families.

the topic is intersting and meets expectations of Plants.

However, The manuscript suffers from many shortcomings.

1-the stucturation needs to be modified deeply for better understanding the developed methodology.

2-authors have cited this sentence "Methods for identifying chromosome ploidy include morphological, cytological, and molecular approaches" why they did not use morphological approach to compare and to highlight the interest of the flow cytometry (both in time and strengh)?

3- the lack of botanical background. Why authors have chosen these three phenological stages?

Moreover, it is not clear the described stages. what authors mean by "Early leaf exhibition". do you mean leaf emergence?  "Leaf forming"? is Leaf expansion ? Mature leaf? is Fully expanded leaf ??

4- what is the added value of the method? this should be discussed and added in conclusion.

Author Response

Response to reviewer

Thank you for your letter and for the reviewers’ comments concerning our manuscript entitled First Insights into Ploidy and Genome Size Estimation in Choerospondias axillaris (Roxb.) B.L.Burtt & A.W.Hill (Anacardiaceae) Using Flow Cytometry and Genome Survey Sequencing (plants-3808733). Those comments are all valuable and very helpful for revising and improving our paper, as well as the important guiding significance to our researches. We have studied comments carefully and have made correction which we hope meet with approval. As required, all revisions have been marked red highlighting in the paper, so that you can see them in Word. In what follows the referees’ comments are in black and the authors’ responses are in red.

#Reviewer 2:

The manuscript deals with the use of new method (technical) to evaluate the ploidy level with a case of study with Choerospondias axillaris.

To do this authors have sampled a several pheological stage of this species and have determined using flow cytrometry the ploidy level and genome size. To ascertain they findings concerning the use of the method they have used other species from the same and different botanical families. the topic is interesting and meets expectations of Plants.

However, the manuscript suffers from many shortcomings.

  1. the structuration needs to be modified deeply for better understanding the developed methodology.

Response: Thank you for your valuable advice. We have deeply revised the developed methodology structuration of the manuscript. The specific modifications include: (1) adding descriptions of the materials used for optimizing flow cytometry (lines 371–373); (2) supplementing Supplementary Tables 1~3 with information such as the number of replicates and CV values under different treatments; (3) revising the processing approach for overlapping bimodal peaks (lines 414–419); and (4) including corresponding methodological details and results related to HiFi sequencing in the Methods section (lines 440–453). The detailed revisions are as follows:

Lines 371–373: The leaves at different stages and floral organs of No. 22 (QYS13) diploid plants collected from the C. axillaris National Germplasm Resources Bank were used to optimize the flow cytometry identification method of C. axillaris (Table S1).

Lines 414–419: Samples exhibiting peak overlap were reanalyzed using a sequential detection protocol. Specifically, both the target samples and the internal standards were independently processed in separate tubes following identical protocols for preparation, propidium iodide (PI) staining, and lysis. Data acquisition was performed sequentially under consistent instrument settings. Genome size and ploidy were subsequently calculated using non-overlapping peak data derived from these independent measurements.

Lines 440–453: The high-quality and purified genomic DNA samples were obtained, and a SART cell sequencing library containing about 15 kb preparation solutions (Pacific Biosciences, CA) cut fragment was constructed. The library preparation involved the following key steps: 1) fragmentation of genomic DNA; 2) repair of DNA damage, end repair, and A-tailing; 3) adapter ligation using the SMRTbell Express Template Prep Kit 2.0 (Pacific Biosciences); 4) nuclease treatment of the SMRTbell library with the SMRTbell Enzyme Cleanup Kit; and 5) size selection and polymerase binding. Sequencing was performed on the PacBio Sequel II platform with Sequencing Primer V2 and the Sequel II Binding Kit 2.0 at the Genome Center of Grandomics. Sequencing data can be found in the National Genomics Data Center under accession number PRJCA031736. A total of 110.6 Gb of HiFi reads were generated and subjected to quality control statistics using SMRTlink, resulting in high-quality valid data. A GC-depth distribution plot was subsequently generated based on these data. The above sequencing was performed at the Wuhan Benagen Technology Co., Ltd.

  1. authors have cited this sentence "Methods for identifying chromosome ploidy include morphological, cytological, and molecular approaches" why they did not use morphological approach to compare and to highlight the interest of the flow cytometry (both in time and strengh)?

Response: Thank you for your constructive suggestion. In fact, we are currently conducting experiments to identify the ploidy of C. axillaris using morphological methods. Initially, due to difficulties in germinating old seeds of C. axillaris, we were unable to obtain suitable materials for chromosome counting. Recently, coinciding with the ripening season of C. axillaris, we have collected seeds of different ploidy levels and are now performing chromosome counting as well as other morphological experiments. These results will be reported in our next article, which will focus on the physiological and morphological differences of polyploid C. axillaris.

  1. the lack of botanical background. Why authors have chosen these three phenological stages?

Moreover, it is not clear the described stages. what authors mean by "Early leaf exhibition". do you mean leaf emergence?  "Leaf forming"? is Leaf expansion ? Mature leaf? is Fully expanded leaf ??

Response: Thank you for your valuable advice and the question you raised is very professional. We selected three distinct phenological stages because C. axillaris is a plant rich in polysaccharides and polyphenols. The accumulation of these compounds during leaf development can interfere with nuclei release and integrity. Therefore, distinguishing leaf developmental stages is critical for optimizing flow cytometry detection in C. axillaris. Regarding the terminology of leaf development stages, the "Early leaf exhibition" is leaf emergence, "Leaf forming" is Leaf expansion, "Mature leaf" is Fully expanded leaf. We have revised the descriptions using more professional terms based on your suggestions.

  1. what is the added value of the method? this should be discussed and added in conclusion.

Response: Thank you for your valuable comment. The added value of this study lies in establishing and optimizing a reliable flow cytometry detection protocol for C. axillaris for the first time, a woody plant rich in secondary metabolites such as polysaccharides and polyphenols. Our protocol successfully addresses typical issues in C. axillaris, including difficulties in nuclei release and high CV values. Compared to conventional methods, our approach identifies WPB as the optimal lysis buffer and young leaves as the most suitable plant material, achieving consistently low CV values ranging from 2.4% to 6.9%. This is well below the generally accepted reliability threshold (CV < 9%), ensuring high accuracy and reproducibility in genome size and ploidy analysis. Most importantly, by applying this method, we have discovered natural triploids in C. axillaris for the first time. This finding reveals previously unrecognized genetic diversity in this species and lays a crucial technical and resource foundation for its polyploid breeding. This discussion has been added to Section 3.1 of the Discussion, and the revised section is as follows:

Lines 257-266: Past research shows that WPB was optimal for flow cytometry analysis in some contexts. For example, WPB performed better in recalcitrant tissues from woody plants [2]. And in Bougainvillea Comm. ex Juss., WPB lysis buffer successfully isolated more intact nuclei, making it the optimal choice for flow cytometry analysis in this species [1]. LB01 buffer was determined to be the most accurate for four bryophyte species (Brachythecium velutinum, Fissidens taxifolius, Hedwigia ciliata, and Thuidium minutulum) [4]. Furthermore, LB01 and Otto's buffers were reported to be the best for young leaf tissue of Sedum burrito, Lycopersicon esculentum, Celtis australis, Pisum sativum, Festuca rothmaleri, and Vicia faba [5]. However, in mature grape leaves, nuclei populations could not be distinguished when using LB01 buffer due to metabolite interference [3].

Lines 279-294: Determining the ploidy of C. axillaris is fundamental for further research on its reproductive development characteristics, as well as for hybridization breeding or ploidy breeding. In our study, ploidy identification was performed for the first time on 58 accessions of C. axillaris germplasm resources. The results showed that the coefficient of variation (CV) values measured by flow cytometry ranged from 2.4% to 6.9%. Previous studies have indicated that flow cytometry results remain reliable when CV values are within 9.0% [36], demonstrating that the pretreatment and detection methods established in this experiment are robust. Based on genome size estimation, 11 samples exhibited a ploidy index between 1.27 and 1.66, with an estimated ploidy level ranging from 2.53 to 3.32, indicating they are triploid. Meanwhile, 47 samples showed a ploidy index between 0.91 and 1.15, with an estimated ploidy level of 1.81 to 2.29, identifying them as diploid. Although no polyploid reports of C. axillaris were found in earlier studies, natural triploids have been documented in other Anacardiaceae species, such as T. vernicifluum [37]. These findings suggest that the C. axillaris germplasm resources assessed in this study exhibit rich genetic diversity based on flow cytometry identification, which can provide diverse parental materials for breeding new varieties.

Round 2

Reviewer 2 Report (New Reviewer)

Comments and Suggestions for Authors

Thank you for the changes made on the manuscript. This latter is now clearer.

This manuscript is a resubmission of an earlier submission. The following is a list of the peer review reports and author responses from that submission.

Round 1

Reviewer 1 Report

Comments and Suggestions for Authors

Suggestions for Improvement

  1. Abstract:
    While informative, it could be slightly condensed to highlight better the key results and implications in a more succinct manner. Consider rewording for clarity and flow. For example:

  2. Introduction:
    The introduction is extensive and informative, but a few redundancies could be trimmed to improve clarity. For instance, the paragraphs on chemical composition and medicinal properties could be streamlined to avoid repeating points already stated.

  3. Results – Flow Cytometry Data:
    Include more descriptive statistical values such as standard deviation for genome size, ploidy frequency distribution, etc ) in the main text to support the visual data in the figures. the figures need more clarity

  4. K-mer Analysis Interpretation:
    The authors acknowledge the discrepancy between flow cytometry and K-mer-based estimates of genome size. It would help to briefly elaborate on how repeat elements or GC content may contribute to this variation beyond technical bias.

  5. Discussion:
    The discussion is sound, but could benefit from:

    More comparative insights: for example, contrasting heterozygosity and repeat content with other members of Anacardiaceae.

    Implications for breeding or conservation: How can this foundational genomic knowledge be translated into practical breeding or conservation strategies?

  6. Figures:
    The figure legends are generally clear, but for key plots such as the K-mer curve, it might help to annotate main peaks and label axes more descriptively to aid reader interpretation.

  7. Minor Language Polishing:
    A professional language editing service or another careful proofreading round is recommended to improve fluency and fix minor grammatical issues: like  “we optimized the ploidy analysis method”  to “we optimized the method for ploidy analysis”.

Recommendation

 I recommend the manuscript for publication in MDPI Plants after minor revisions focused on improving clarity, polishing the language, and expanding a bit on the discussion's implications. The content is strong, relevant, and supported by appropriate methodology and literature.

Comments on the Quality of English Language

Need to be improved

Reviewer 2 Report

Comments and Suggestions for Authors

Choerospondias axillaris is an economically important fruit-producing tree in Asia. Studies of its genome are of course very important. However, I see some important problems in the present manuscript.

The plant belongs to the family Anacardiaceae that also includes other very important cultivated plants with a lot of genomic work performed. Neither the title not the abstract indicates the family placement of the plant. The first time we read about Anacardiaceae is the following phrase: “The top five species were all from Anacardiaceae family species except Ailanthus altissimus (Table 3). The species with the highest sequence alignment rate was the chloroplast genome of Choerospondias axillaris (3.01%), followed by Pistacia vera (2.5%), indicating that Pistacia vera was closely related to Choerospondias axillaris.” – While reading this, a reader still does not know whether Choerospondias taxonomically belongs to Anacardiaceae. Furthermore, when I see that Ailanthus is more similar to Choerospondias than two members of Anacadriaceae in the analysis of authors’ data, I conclude that these data are far from being representative enough. Why plastid genome was of interest here if the paper is related to polyploidy that is determined by the nuclear genome?

The authors provide interesting data on the occurrence of diploid and triploid plants using flow cytometry. However, instead of presenting a table with flow cytometry statistics, they show images of fresh data with peaks. The supplementary table provides information about the accessions used. However, it lacks two important parameters: (1) whether the accessions originated from wild populations or cultivated/domesticated plants and (2) which plants were male and which were female. Choerospondias is a dioecious plant, and comparing genome sizes of males and female will be essential. All the work on nuclear sequences should be clearly linked to the gender used. Is there any geographical preference of triploids? Do triploid plants set fruits? If yes, are these fruits parthenocarpic? What about pollen sterility? Is there any morphological character correlating with ploidy level?